# The Diverging Routes of BORIS and CTCF: An Interactomic and Phylogenomic Analysis

**DOI:** 10.3390/life8010004

**Published:** 2018-01-30

**Authors:** Kamel Jabbari, Peter Heger, Ranu Sharma, Thomas Wiehe

**Affiliations:** Cologne Biocenter, Institute for Genetics, University of Cologne, Zülpicher Straße 47a, 50674 Köln, Germany; peter.heger@uni-koeln.de (P.H.); ranusharma09@gmail.com (R.S.); twiehe@uni-koeln.de (T.W.)

**Keywords:** CTCF, gene duplication, chromatin loops, polymorphism, natural selection, Bilateria, Amniotes

## Abstract

The CCCTC-binding factor (CTCF) is multi-functional, ubiquitously expressed, and highly conserved from *Drosophila* to human. It has important roles in transcriptional insulation and the formation of a high-dimensional chromatin structure. CTCF has a paralog called “Brother of Regulator of Imprinted Sites” (BORIS) or “CTCF-like” (CTCFL). It binds DNA at sites similar to those of CTCF. However, the expression profiles of the two proteins are quite different. We investigated the evolutionary trajectories of the two proteins after the duplication event using a phylogenomic and interactomic approach. We find that CTCF has 52 direct interaction partners while CTCFL only has 19. Almost all interactors already existed before the emergence of CTCF and CTCFL. The unique secondary loss of CTCF from several nematodes is paralleled by a loss of two of its interactors, the polycomb repressive complex subunit SuZ12 and the multifunctional transcription factor TYY1. In contrast to earlier studies reporting the absence of BORIS from birds, we present evidence for a multigene synteny block containing CTCFL that is conserved in mammals, reptiles, and several species of birds, indicating that not the entire lineage of birds experienced a loss of CTCFL. Within this synteny block, BORIS and its genomic neighbors seem to be partitioned into two nested chromatin loops. The high expression of SPO11, RAE1, RBM38, and PMEPA1 in male tissues suggests a possible link between CTCFL, meiotic recombination, and fertility-associated phenotypes. Using the 65,700 exomes and the 1000 genomes data, we observed a higher number of intergenic, non-synonymous, and loss-of-function mutations in CTCFL than in CTCF, suggesting a reduced strength of purifying selection, perhaps due to less functional constraint.

## 1. Introduction

The CCCTC-binding factor CTCF plays a critical role in transcriptional regulation in vertebrates (for reviews, see [1]). It was first identified by its ability to bind different regulatory sequences in the promoter-proximal regions of the chicken, mouse, and human MYC oncogene [2,3]. CTCF is an 11 C2H2 zinc finger nuclear protein and is involved in enhancer blocking, gene imprinting, nucleosome positioning, chromatin remodeling, and promoter activation/repression [4]. Together with cohesin, CTCF mediates chromatin folding and stabilizes chromatin loops. CTCF knockout experiments in mice proved its crucial role in development ([5] for a review). Deregulation of CTCF has been linked to cancer in a broad range of tissues [6].

A gene duplication in the ancestor of amniotes generated the CTCF paralog CTCFL (CTCF-like), also called “Brother of Regulator of Imprinted Sites” (BORIS) in humans [7]. BORIS is known to be de-regulated in cancer [8,9]. Its disruption in mice causes sub-fertility because of a partially penetrant testicular atrophy. BORIS knockout mice homozygous for the null allele had a defect in spermatogenesis that resulted in small testes associated with an increased cell death [10]. CTCFL deficiency affects the expression of a number of testis-specific genes [11]. It has a DNA binding specificity similar to that of CTCF, but divergent N- and C-termini (see Figure 1) [12]. 

Even if CTCF and CTCFL bind similar DNA motifs, only two-thirds of the CTCFL-binding sites are bound by CTCF, and only a subset (~29–38%) of CTCF binding regions were also occupied by BORIS [11]. It has been reported that CTCF and BORIS bind competitively to common sites and display opposing effects on the epigenetic status of the Igf2/H19 imprinting control region and on transcription of the BAG1 and CT-genes [13,14,15]. Thus, after the duplication event, CTCF and BORIS must have undergone sub/neo-functionalization [16] to take on tissue-specific roles. Indeed, in mammals, CTCFL appears to be expressed primarily in spermatogonia and preleptotene spermatocytes [17], while CTCF is expressed in the nucleus of somatic cells. During male germ cell differentiation, CTCF and CTCFL proteins are normally expressed in a mutually exclusive pattern that correlates with the resetting of histone methylation marks [13]. RT-PCR experiments showed two main patterns of BORIS expression. In marsupials and eutherian mammals (wallaby and cattle, respectively), it is found predominantly in testes with slight ovarian expression, whereas in the reptile bearded dragon and the monotreme platypus, the expression of BORIS was detected in multiple somatic tissues as well as in the gonads, indicating that BORIS may have had a wide expression in ancestral amniotes, similar to that of CTCF, the gene from which it arose by duplication [7].

To better understand the functional divergence of CTCF and CTCFL, we investigated their protein interaction networks from an evolutionary point of view and analyzed the genomic conservation of orthologous gene order (synteny) in their neighborhood. Chromatin conformation capture data, in particular sperm Hi-C data, adds another dimension to our investigation, establishing a possible link between chromatin topology within the CTCFL synteny block and common gene expression patterns.

## 2. Materials and Methods

### 2.1. Construction of Protein Interaction Networks

The protein interaction networks of CTCF and CTCFL were built by importing the protein-encoding gene symbols of the HUGO gene nomenclature committee (HGNC) into the Cytoscape 3.4.0 [17] plugin BisoGenet 3.0.0 9 [18] with the BioRelation type ‘protein-protein interaction’ selected.

### 2.2. Annotation of Network Proteins

Gene set enrichment analysis (GSEA) of CTCF/CTCFL interaction partners was carried out by DAVID, the Database for Annotation, Visualization and Integrated Discovery [19]. DAVID has a high integration level of 40 functional annotation categories from dozens of heterogeneous public databases. Results from DAVID were summarized and visualized using REViGO (reduce + visualize Gene Ontology). REViGO is a web-based server that condenses long, complex lists of GO terms by detecting a bona fide subset of the GO terms through clustering based on semantic similarity quantification. REViGO also helps in the interpretation of the set of non-redundant GO terms (see Table 1). The choice of the groups’ elements is facilitated by the *p*-values and GO terms that the user provides. If the *p*-values are very similar and one term is a child node of the other, REViGO will prefer to select the parent term. In the present work, we used GO-terms with a *p*-value of less than 0.05 and count numbers larger than 2 [20,21].

### 2.3. Phylogenomics of CTCF and BORIS

We previously constructed a large-scale orthology database of opisthokont sequences, based on annotated genomes, proteomes, and conceptually translated ORFs (PH, unpublished results). This database allowed us to determine the phylostratigraphic ages of all interaction partners collected above and to investigate the loss of CTCF interactors in the nematode phylum. 

The structure, size, and gene content of syntenic regions were investigated using the GENOMICUS [22,23] database, which also offers a framework to examine reconstructed ancestral ortholog clusters and a flexible visualization interface. The Ensemble database (http://www.ensembl.org) was used to search for CTCFL interaction partners in chicken and other birds. Information on bird CTCFLs collected from the NCBI nucleotide database, including those represented in the phylogenetic tree (see Figure 2), is reported in Appendix A.

The sequence dataset used for phylogenetic analysis consisted of 40 vertebrate CTCF/CTCFL protein sequences downloaded from www.uniprot.org and NCBI (bird CTCFLs). Multiple sequence alignments were created using the MAFFT v7.312 “EINSI” algorithm [24]. Indels and unalignable regions were excluded manually from the data prior to analysis. Phylogenetic trees were computed under the maximum likelihood criterion, using RAXML 8.1.3 with 50 distinct randomized MP trees and 100 bootstrap resamplings under the LG+Gamma model of sequence evolution [25]. Phylogenetic trees were edited with TreeGraph [26] and Affinity Designer Version 1.6 (https://affinity.serif.com). 

### 2.4. Analysis of Chromatin Loops

To examine genome-wide chromatin interaction frequencies, Hi-C maps were visualized using Juicebox [27], a software for plotting data from proximity mapping experiments. Hi-C maps were visualized at different resolutions. For confirmation purposes, we used two independent Hi-C datasets from mouse sperm [28,29]. The sequencing results of Hi-C libraries of sperm cells and fibroblasts from [28] are available at the NCBI Sequence Read Archive under accession number SUB540202 (SRX553176). The accession number for the Hi-C dataset reported in [29] is GEO: GSE79230.

### 2.5. CTCF and CTCFL Tolerance to Mutational Changes

To analyze genetic variation in the protein-coding part of genes, we used the exome database (ExAC) containing data from 60,706 humans, collected worldwide [30,31]. To assess the deviation of observed mutation counts from the expected number, a signed Z score is provided. Positive Z scores indicate increased constraint (intolerance to variation) and imply that a gene had fewer variants than expected. Negative Z scores designate genes with more variants than expected.

The expected and observed variant counts are used to calculate the probability that a given gene is intolerant to loss-of-function (LoF) variation. Three classes of genes are defined based on their tolerance to LoF variation: null if LoF variation is fully tolerated, recessive if heterozygous LoFs are tolerated, and haploinsufficient if heterozygous LoFs are not tolerated. Intolerance to loss of function (pLI) is high if its value is closer to 1, and the gene is considered intolerant to LoF (for details see [30]).

For the 1000 genomes data, we used the ratio of the normalized number of mutations in CTCFL and CTCF (r = N_CTCFL_/N_CTCF_) as a relative measure of variability. Stop codon gain, frameshift, missense, and synonymous variant counts were divided by the length of the coding sequence of CTCF (2181 bp) and CTCFL (1989 bp). Intron variants were normalized by dividing the number of mutations by the respective total intron sizes (74,595 bp for CTCF and 27,697 bp for CTCFL). UTR mutations were not normalized because they are of a fixed size (2 or 5 kb).

## 3. Results

### 3.1. Interaction Networks of CTCF and CTCFL Are Diverse and Overlap Poorly

CTCF is a multifunctional protein and interacts with several other factors to carry out these functions, thereby creating a protein interaction network with these cofactors. To examine the consequences of the CTCF duplication on cofactor network topology, we analyzed the protein interaction partners of both duplicates, CTCF and CTCFL. The first order interaction network of CTCF revealed 52 interaction partners. Out of (*52 choose 2*) = 1326 combinatorially possible pairwise interactions, 253 interactions (19.1%; excluding dimers) are experimentally confirmed, as reported by Cytoscape; 52 of them are between CTCF and its partners; and 201 network edges are among the partners without the participation of CTCF (Figure 2A). After CTCF, the network nodes with the highest degree of connection are UBC (degree 40) and SUMO2 (degree 30); both are involved in post-translational modification of CTCF. Notice that interactions missing in Cytoscape (CTCF-BORIS [14] and BORIS-TAF7L [32]) are not considered here, but they are included in Figure 3.

Like CTCF, CTCFL has a central domain with 11 C2H2 zinc fingers (Figure 1) and binds almost the same consensus DNA sequences as CTCF [10]. However, the CTCFL interaction network differs substantially in size from that of CTCF (Figure 2B). Excluding homodimers, there are 19 first-tier interaction partners of CTCFL (only 17 of them are shown in Figure 2B), and only three proteins are shared between both PPI networks, the heat shock protein beta-8 (H11), the histone H2A type 2-C (H2AC), and BORIS. After CTCFL, H3.1 is the most connected protein in the BORIS PPI network. In contrast to histone H3.3, which is only incorporated outside the DNA replication phase [33], the CTCFL partner histone H3.1 is assembled into nucleosomes during DNA replication. There is a widespread reduction in H3.1 nucleosome occupancy around transcriptional start sites in round spermatids compared to mature sperm where the canonical histones H3.1/H3.2 are retained [34].

Although different in size, the two networks have a similar average adjacency (19.1% for CTCF and 20.3% for CTCFL) and may still be involved in common cellular pathways. To investigate this possibility, we performed a gene ontology enrichment analysis (see Table 1). 

The GO-term enrichment analysis for the interaction partners of both CTCF and CTCFL illustrates the mechanisms and pathways which may affect and regulate CTCF/L function in development or during aberrant processes such as tumorigenesis. Querying the GO-database for overrepresented terms, we found “Nucleoplasm” as the most enriched term for both CTCF and CTCFL (*p*-value = 2.1 × 10^−15^ and 2.1 × 10^−9^, respectively), reflecting their cellular location as chromatin agents. The next overrepresented GO terms reflect more divergent categories, “Positive and negative transcription regulation” for CTCF (*p*-value = 2.1 × 10^−9^) and “Hsp70 binding” and “Protein folding” for CTCFL (*p*-value = 4.4 × 10^−4^ and 9.4 × 10^−2^).

### 3.2. Phylogenomic Distribution of CTCF/CTCFL and Their Interaction Partners

To investigate whether the interaction networks of CTCF and BORIS evolved successively after the origin of the two proteins, we determined the phylostratigraphic ages of all interaction partners. As summarized in Figure 3, many of the CTCF and BORIS interactors are evolutionarily old and originated in the ancestor of opisthokonts or earlier. Almost all interactors pre-date the origin of CTCF (45 out of 52; 87%) and BORIS (17 out of 19; 89%) themselves. From the human perspective, it thus appears that both protein-protein interaction networks were completely established shortly after the origin of CTCF/CTCFL and are evolutionarily stable since hundreds of millions of years, with only minor additions or losses at later times. 

IFI16 (γ-interferon-inducible protein 16) inflammasome, HSB9 (Heat shock protein beta-9), and H11 (Heat shock protein b8) are the youngest interactors of CTCF. IFI16 has been shown to form inflammasomes that respond to DNA molecules in the nucleus [35]. It also senses DNA forms of the lentiviral replication cycle and controls HIV-1 replication [36]. If the sequence sensors of IFI16 have a differential affinity to short sequences similar to CTCF binding sites, this property may be a facilitator for direct or indirect interactions between IFI16 and CTCF. In line with this possibility is the presence of ectopic CTCF binding sites in the Human T-Lymphotropic Virus type 1 (HTLV-1), which causes malignant and inflammatory diseases in ∼10% of infected people [37]. Not much is known about the link of CTCF to inflammasomes except that CTCF depletion in mice brain leads to the over-expression of inflammation-related genes and microglial dysfunction [38].

### 3.3. Parallel Loss of CTCF/CTCFL and Their Interaction Partners

Because CTCF was lost secondarily in some nematode clades [39], we interrogated our database for possible losses of its interactors. Five out of 52 CTCF interactors were lost together with CTCF during nematode evolution (see Appendix A). They include (i) RXRA, transcriptional co-repressors that induce histone acetylation, chromatin condensation and transcriptional suppression; (ii) SUZ12, a Polycomb group (PcG) protein and component of the PRC2/EED-EZH2 complex; (iii) YY1, a multifunctional transcription factor that exhibits positive and negative control on a large number of cellular and viral genes by binding to sites overlapping the transcription start site. YY1 was proposed to recruit the PRC2/EED-EZH2 complex to transcriptionally repressed target genes; and (iv) ZMYM2 plus ZMYM4, the former being a zinc finger protein that may be a component of a histone deacetylase complex, the latter of which is able to inhibit interferon-gamma induced apoptosis. Loss of all five interactors is a peculiar feature of members of the derived nematode clade V, namely, Rhabditida and Strongylida (e.g., *C. elegans*, *C. briggsae*, *Haemonchus contortus*, *Ancylostoma caninum*). Ancestral clades, such as Mermithida and Trichocephalida (e.g., *Trichuris* and *Trichinella*), are less affected by gene loss, as one may expect from the fact that they possess CTCF. Whether the observed correlation between the loss of CTCF and some of its interaction partners in nematodes is causally related or reflects the large differences between the two nematode subgroups Enoplea and Chromadorea needs to be determined in future experiments. 

Chicken appears to be the only known amniote for which CTCFL was lost secondarily, but no interaction partner was specifically lost in this lineage, except BAG6, which was also lost in other birds that possess CTCFL. We should mention here that the partially inbred, red jungle fowl serves as the chicken reference genome [40] and does not necessarily represent the entire genus or species. Moreover, the absence of experimental evidence for CTCFL loss in this inbred species casts still doubt on this finding.

A clear example of how assembly issues and uncompleted genome sequencing data may be misleading is our observation that SPO11 seems to be missing in the corresponding genomic region of the lizard. Spo11 may not be sequenced yet or may lay in an un-assigned sequence of the genome. Indeed, a homology search on DNA sequences from *Anolis* showed significant similarity to an annotated ORF (see Appendix A) that contains InterPro domain IPR036078, characteristic of the superfamily: Spo11/DNA topoisomerase VI subunit A. Blast search of this ORF in the NCBI nucleotide sequence database retrieved SPO11 homologues from all classes of vertebrates, including other reptiles. It is therefore likely that in the Anolis lizard, Spo11 is miss-assigned due to assembly issues (highly gapped reference genome), and a further hint to this suspicion stems from our finding of larger contigs from *Alligator mississippiensis* and *Crocodylus porosus* (see Appendix A) with conserved synteny in other birds and mammals.

### 3.4. BORIS is Lost in Chicken, But not from All Birds

Previous searches for BORIS [7] orthologs in birds detected only a fragment of 160 bp with strong similarity to CTCFL. As this sequence was located between two genes in synteny with other mammalian species, the authors concluded that either BORIS experienced pseudogenization in birds after they diverged from reptiles, or underwent a rapid functional change leaving behind only small traces of its evolutionary past. In agreement with the latter scenario, the new version of the chicken genome assembly (Gallus_gallus-5.0; GCA_000002315.3; [40]) did not recover CTCFL. This does not exclude that CTCFL is still in the un-assembled part of the genome. Unexpectedly, when we did a NCBI homology search, we found several BORIS gene models (Appendix A) with up to 10 zinc-finger domains in birds (e.g., in *Meleagris gallopavo*, *Taeniopygia guttata*, *Numida meleagris*, *Parus major*, *Pseudopodoces humilis*, *Sturnus vulgaris*, *Ficedula albicollis*). To test if they represent *bona fide* CTCFL orthologs, we performed maximum likelihood phylogenetic analysis using amino-acid sequences of CTCF and CTCFL from different vertebrate species. The resulting tree (Figure 4) shows that all bird CTCFLs cluster with their mammalian and reptilian orthologs. The CTCFL sub-tree has longer branches compared to the CTCF clade, and this signature is expected under relaxed selection on the derived copy compared to the ancestral gene. 

Bird CTCFLs not only exhibit strong similarity to other amniote orthologs, but also feature conserved synteny with several surrounding genes (Figure 5), including SPO11 (initiator of meiotic double strand breaks), RAE1 (ribonucleic acid export 1), RBM38 (RNA binding motif protein 38), PCK1 (phosphoenolpyruvate carboxykinase 1), ZBP1 (Z-DNA binding protein 1), and PMEPA1 (prostate transmembrane protein, androgen induced 1) (for more details, see Appendix A).

These genes are located within two nested loops in mouse sperm cells and may represent an insulated region with high self interaction (see Figure 6 and Appendix A). The smallest sub-loop harbors SPO11, RAE1, RBM38, and CTCFL; all four genes are highly expressed in sperm cells (RBM38 is widely expressed). The larger loop contains PCK1, ZBP1, and PMEA1. Two of its genes are lowly expressed in sperm cells, but the third gene, PMEA1, is highly expressed in male tissues. PCK1 and ZBP1 seem to be under repressive effects of a lamina associated domain (LAD) that is marked by the typical GC-poor signature of LAD and that was observed in several tissues; to our knowledge, no data on LADs maps is available for sperm cells (see Appendix A). It is not clear how the conformational state in this chromatin area can bring about the set of enhancers and promoters in the neighborhood. Our speculation about a common regulatory environment in this genomic area will need experimental validation, in particular in sperm cells, where CTCFL is expressed.

### 3.5. CTCF Is Less Mutation Tolerant than BORIS

Another approach to evaluate the propensity of CTCF/CTCFL to mutate can be gleaned from human population data. By knowing the spectrum of mutations (single nucleotide and copy number variations) of each gene, one may plausibly extrapolate the observed features to interspecies divergence. 

Querying the Exome Aggregation Consortium (ExAC) database [30], we noticed that CTCF has increased mutational constraints (intolerance to variation) compared to BORIS (Table 2). The intolerance to loss-of-function, pLI, is 1 for CTCF, making it strongly intolerant to such mutations. We may point out here that PRDM9, a chromatin regulator involved in meiotic recombination and similar to CTCF in structure (8 zinc-fingers) and binding site affinity, has a pLI value of 0, indicating its propensity to mutate and ultimately lose function, as is the case for birds [41] and dogs [42]. Taking into consideration CTCF interactors that were lost during nematode evolution, two of them, SUZ12 and ZMYM4, are highly intolerant to deletions according to the human exome population data (Table 2), as their pLI values are equal to or even higher than 1. 

Using the 1000 genomes variation data (26 surveyed populations), the pattern of higher variation in CTCFL compared to CTCF was also confirmed for the non-coding parts of the genes (Figure 7). 

## 4. Discussion

In this study, we compared protein interaction networks of CTCF and BORIS and identified an inflammasome component (IFI16), the small heat shock protein HSPB9, and H11 (heat shock protein beta-8) as the evolutionarily youngest interactors of CTCF and BORIS; H11 and H2AC (histone H2A type 2-C) are the only common interaction partners of CTCF/CTCFL and originated in the ancestor of mammals. All other CTCFL and CTCF interactions evolved prior to the origin of amniotes. The restricted CTCFL expression in male tissues (mainly in sperm cells) may in part explain the differences between the CTCF and CTCFL PPI networks. If so, more similarities between the interaction networks would be expected in reptiles or monotremes, where the expression of CTCFL is less restricted [7]. The co-expression of CTCFL and CTCF in late spermatogonia and preleptotene spermatocytes poses the question whether they are redundant, complementary, or antagonistic in function. The dissimilarity of the PPI networks indicates functional divergence rather than redundancy. This is also supported by gene replacement experiments in transgenic mice [11]. Antagonistic relations are unlikely, not only because of sequence divergence of the two proteins, but also because of a change in the chromatin context following duplication and translocation [43,44], leading to tissue specificity and limited competition between CTCF and CTCFL on DNA binding sites. Moreover, the physical CTCF-BORIS interaction in germ cells is an argument in favor of cooperative rather than antagonistic relations [11].

Phylostratigraphy reveals that the majority of interacting proteins pre-existed and possibly did not co-evolve with CTCF/CTCFL. However, it is not clear how and when the interactions themselves were formed/eliminated. Did the interactions form early after the appearance of CTCF or CTCFL, or only gradually over a longer evolutionary time span? For comparison, more than 100 interactions may be added to the yeast proteome network every million years, some of which add previously unconnected proteins to the network [45]. Assuming that new interactions are formed with a more or less constant rate, one would indeed expect a much larger number of interactors for CTCF than for CTCFL due to its more ancient origin. However, only very few connections to younger proteins (see Figure 3) have been established (HSPB9, H11, IFI16, and CTCFL). The birth of CTCFL occurred later in vertebrate evolution and may have had equal chances of interacting with all potential CTCF interactors. However, if the translocation of the new copy occurred early after the duplication event took place (~260 MYA), the newly acquired chromatin context of CTCFL is likely to shield this copy not only from interference with CTCF, but also from the interaction with its previous partners. The data from Figure 3 shows that little overlap exists between the two protein networks, although CTCF and CTCFL are both expressed in male tissues and thus exposed to the same protein pool, indicating that in addition to the genomic context, mutations that changed the protein surface may have contributed to the divergence of protein interactions.

Examination of syntenic regions around vertebrate CTCF and CTCFL led us to notice that many CTCFLs, thought to be lost in birds, are actually present and that their synteny with other mammals and reptiles is still conserved. Conservation of gene order around CTCFL is higher compared to CTCF, illustrating the antiquity of the CTCF neighborhood (paleo-synteny), which originated in the bilaterian ancestor before the divergence of arthropods and vertebrates [46]. The genes in the chromosomal neighborhood of CTCFL, born at least before the emergence of amniotes, constitute what we call a “neo-synteny”. Purifying selection appears to be relaxed in CTCFL (see branch length in Figure 4) compared to CTCF. 

Incidentally, when preparing this manuscript, a new study [47] proposed that the CTCF duplication occurred earlier in vertebrate evolution than previously suggested, that is, before the split of the chondrichthyan lineage and not in the ancestor of amniotes. This conclusion rests on phylogenetic arguments, based on a CTCF duplication specific to lamprey. As no shared CTCFL has been identified in species other than amniotes, we are not convinced that this interpretation is correct. However, it cannot be strictly excluded that an ancient CTCFL paralog may have been lost multiple times independently in early vertebrate evolution.

The secondary loss of CTCF in derived nematode clades is accompanied by the loss of key interactors such as the Polycomb group (PcG) protein Suz12 and the multifunctional transcription factor TYY1, reflecting the strong link between the complexity of epigenetic modifications (protein and DNA) and CTCF function. It is unclear how CTCF can dispose of some interaction partners; the best candidates for such elimination will be one-way interactors, those that interact with CTCF and not, or only loosely, with other proteins. If so, the potential burden caused by loss of function changes will be tolerated because it will not be detrimental to other sub-hubs connected to it. This implies that some kind of selection can operate on the network as a unit. Evidence in this direction is available for the human interactome as a whole [48] and for disease susceptibility loci in particular [49].

The number and distribution of variants obtained from the 1000 genomes and the 64,000 exomes data indicate a relaxed constraint on all gene parts of CTCFL. This is expected not to alter the expression pattern alone, but also the binding affinity. Despite highly conserved synteny around CTCFL, this gene shows by far higher mutability and consequently a higher tolerance for mutational burden and even for loss of function compared to CTCF. The ancestral copy (CTCF) also has a wide tissue expression, whereas the derived one (CTCFL) is expressed only in mammalian male tissues (sperm and testis), together with its neighboring genes, in particular SPO11 (initiator of meiotic double stranded breaks) and PMEPA1 (prostate transmembrane protein, androgen induced 1). SPO11 expression is limited to male tissues like CTCFL; the combined expression of these genes in sperm cells is reminiscent of reduced fertility phenotypes associated with CTCFL deficiency in mice [10,11] and prostate cancer [50,51]. Together, these data indicate that co-regulation may be critical for the genes located in this genomic area, as expected from the observation that these genes belong to nested chromatin loops (see Figure 6 and Appendix A). We hypothesize that this chromatin conformation may point to a complex regulatory unit that brings together CTCFL, meiotic recombination, and possibly a fertility associated phenotype, which may have consequences for evolutionary fitness.

## Figures and Tables

**Figure 1 life-08-00004-f001:**
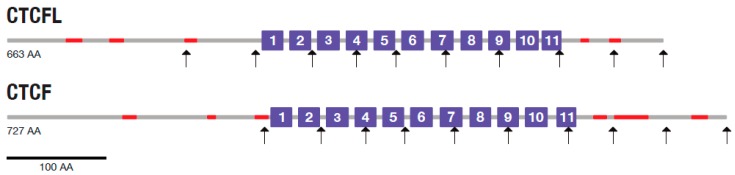
Structural comparison of CTCF and CTCFL. Numbered boxes indicate C2H2 zinc-finger domains, vertical arrows demarcate the position of introns, and red bars represent low complexity regions. Drawn to scale.

**Figure 2 life-08-00004-f002:**
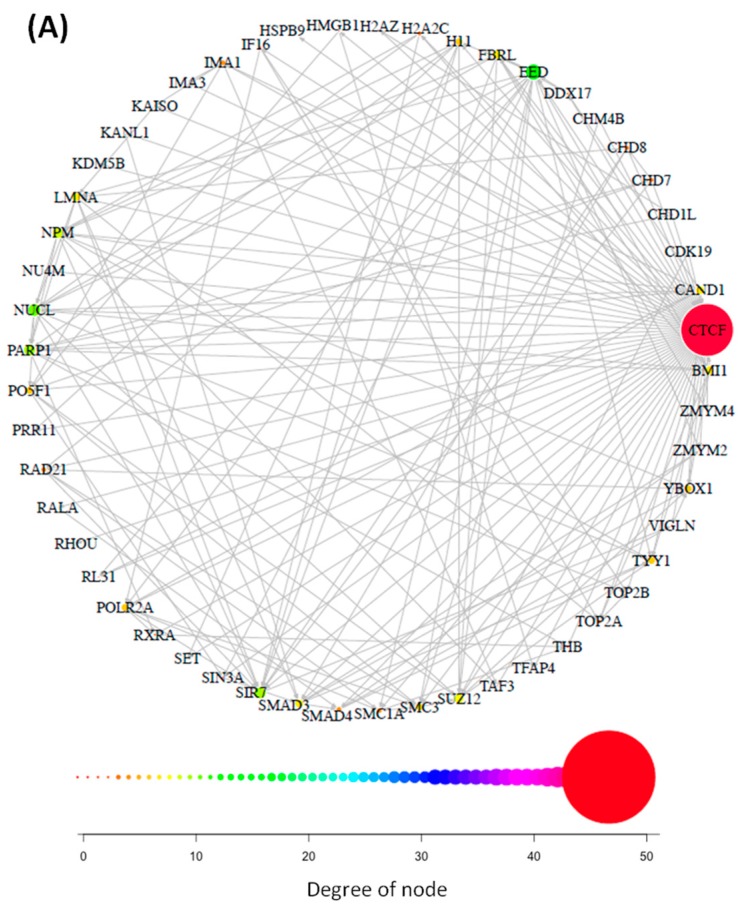
Protein interaction network of CTCF (**A**) and CTCFL (**B**): Colour (from orange to red) and node size denote increasing degree.

**Figure 3 life-08-00004-f003:**
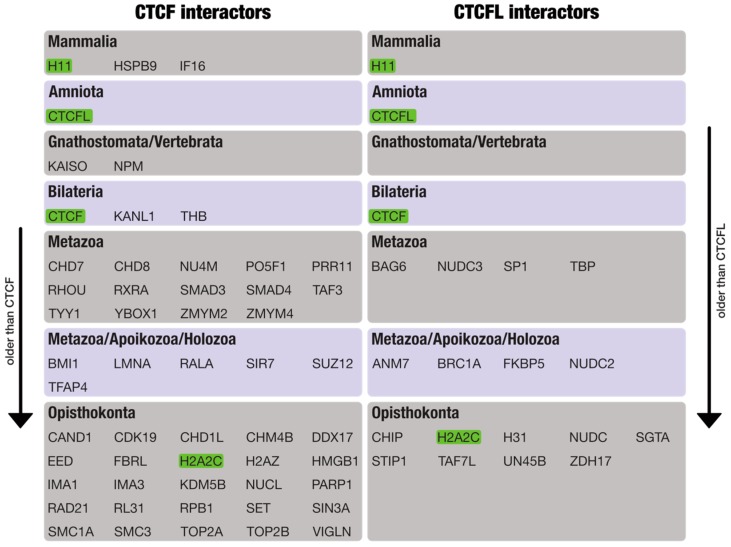
Lineage phylostratigraphy of CTCF/CTCFL interactors. Note that Apoikozoa is the common ancestor of animals and choanoflagellates and Holozoa is the common ancestor of Apoikozoa and Ichthyosporea (Mesomycetozoea). Multiple lineage assignments indicate that a particular ancestor cannot be inferred with certainty. Assignment to Opisthokonta indicates that the protein emerged in the ancestor of Opisthokonta or earlier. Green refers to common interactors, arrows on the margins indicate increasing evolutionary age relative to CTCF(L).

**Figure 4 life-08-00004-f004:**
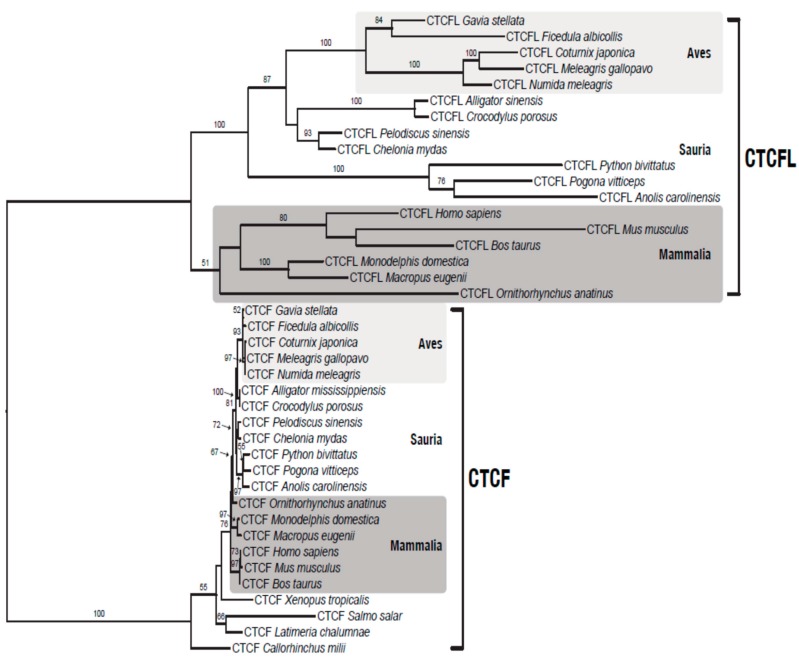
Maximum likelihood phylogeny of the CTCF gene family. The underlying alignment matrix consists of 40 protein sequences with 772 characters and 15.56% gaps or undetermined characters. The main aim of this tree is to show that CTCFs and CTCFLs from birds form different clusters. Fish CTCFs have atypical phylogenetic positions in the tree, as reflected by their low bootstrap values. Bootstrap values below 50 were omitted for clarity.

**Figure 5 life-08-00004-f005:**
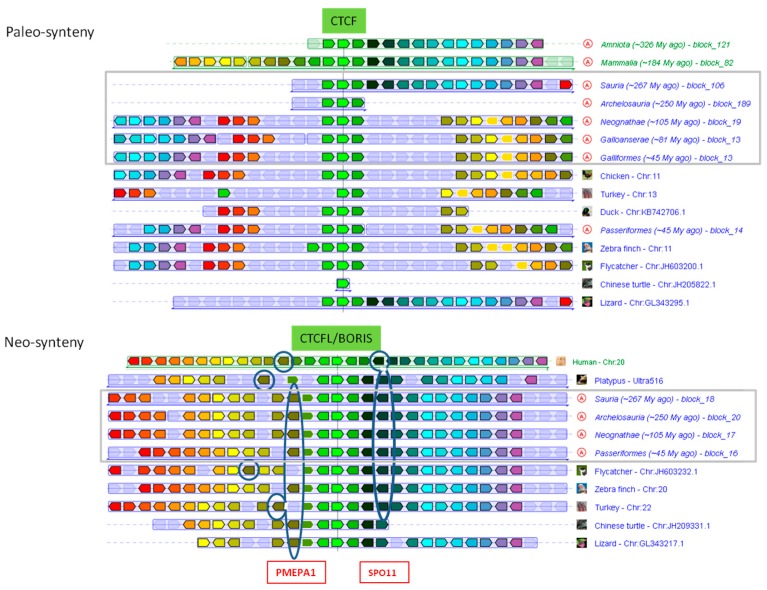
Genomic synteny blocks of CTCF/CTCFL: Boxes (transcript orientation) with identical colors correspond to orthologous genes. Encircled “A” in the grey frame corresponds to the inferred ancestral state (see *Genomicus* web site). Neo-synteny/Paleo-synteny refers to evolutionary younger/older orthologous clusters.

**Figure 6 life-08-00004-f006:**
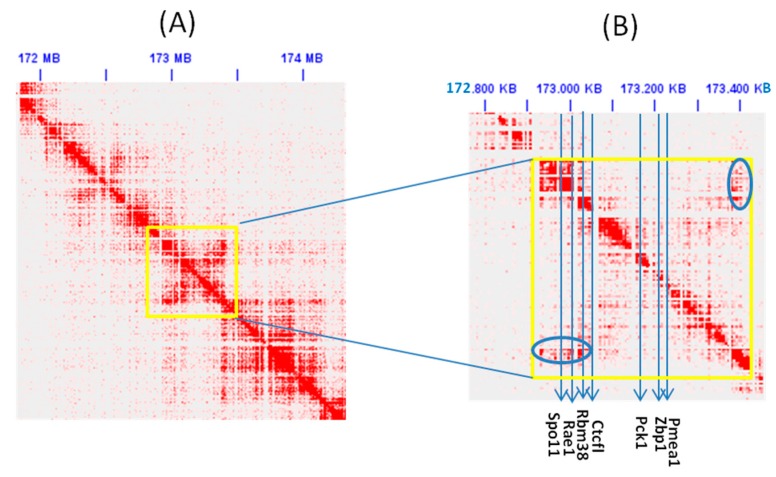
Hi-C map of mouse sperm cells at a 25 kb resolution showing the chromatin (sub) loop harboring SPO11, RBM38, and CTCFL, and a larger loop containing PCK1, ZBP1, and PMEA1. The region of interest (Chr2:172,970,000-173,224,000) is marked with a yellow frame (**A**) and zoomed in at a 5 kb resolution (**B**). Green circles mark off-diagonal interactions.

**Figure 7 life-08-00004-f007:**
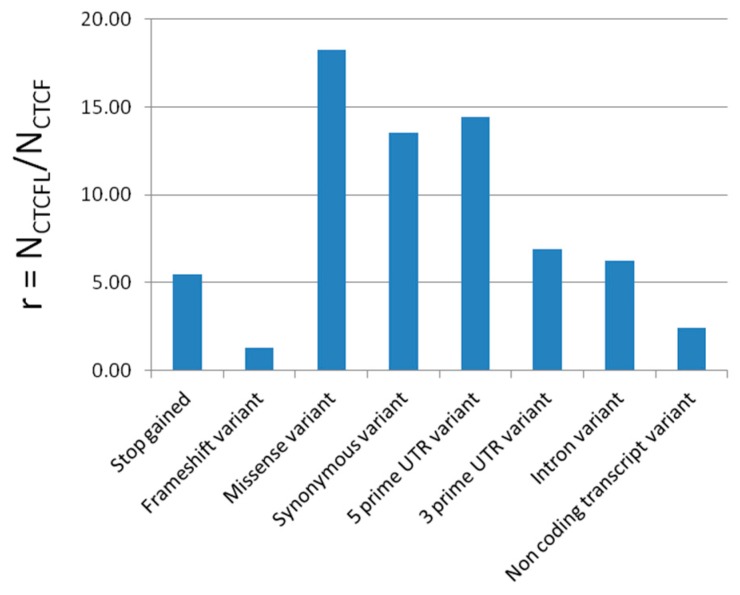
Histogram showing enrichment for mutations in CTCF(L) based on the 1000 genomes phase 3 data. The y-axis displays the ratio of the normalized number of mutations in CTCFL and CTCF (r = N_CTCFL_/N_CTCF_).

**Table 1 life-08-00004-t001:** GO-term enrichment analysis for CTCF(L) interactors.

CTCFL GO-Term	GO-Term Name	*p*-Value
GO:0005654	Nucleoplasm	1.80 × 10^−4^
GO:0030544	Hsp70 protein binding	4.40 × 10^−4^
GO:0006457	Protein folding	9.40 × 10^−2^
GO:0005737	Cytoplasm	2.60 × 10^−2^
GO:0005515	Protein binding	1.20 × 10^−1^
GO:0051082	Unfolded protein binding	1.10 × 10^−1^
GO:0051787	Misfolded protein binding	1.90 × 10^−1^
GO:0000790	Nuclear chromatin	1.90 × 10^−1^
GO:0006359	Regulation of transcription from RNA POL-III promoter	6.30 × 10^−1^
GO:0006349	Regulation of gene expression by genetic imprinting	5.60 × 10^−1^
**CTCF GO-Term**	**GO-Term Name**	***p*****-Value**
GO:0005654	Nucleoplasm	2.10 × 10^−15^
GO:0005634	Nucleus	1.30 × 10^−14^
GO:0045944	Positive regulation of transcription from RNA POL-II promoter	2.10 × 10^−9^
GO:0000122	Negative regulation of transcription from RNA POL-II promoter	3.00 × 10^−9^
GO:0005515	Protein binding	2.00 × 10^−8^
GO:0003677	DNA binding	5.90 × 10^−7^
GO:0001701	In utero embryonic development	6.00 × 10^−7^
GO:0044822	Poly(A) RNA binding	7.80 × 10^−7^
GO:0005730	Nucleolus	9.60 × 10^−7^
GO:0046982	Protein heterodimerization activity	3.10 × 10^−6^

**Table 2 life-08-00004-t002:** CTCF/CTCFL tolerance to mutational burden as reflected in the ExAC data base on 60,706 exomes.

	Constraint	Expected	Observed	Constraint
	from ExAC	No. Variants	No. Variants	Metric
**CTCFL**	Synonymous	106.7	116	z = −0.56
	Missense	248.7	223	z = 0.80
	LoF	22.3	4	pLI = 0.78
	CNV	8.1	1	z = 1.19
**CTCF**	Synonymous	102.4	123	z = −1.26
	Missense	274.9	110	z = 4.86
	LoF	22.2	1	pLI = 1.00
	CNV	6.4	2	z = 0.75
**RXRA**	Synonymous	111.3	112	z = −0.04
	Missense	217.5	114	z = 3.43
	LoF	13.2	1	pLI = 0.94
	CNV	6.2	4	z = 0.34
**SUZ12**	Synonymous	66.7	54	z = 0.96
	Missense	181.4	80	z = 3.68
	LoF	25	1	pLI = 1.00
	CNV	nan	nan	z = nan
**YY1**	Synonymous	118.4	83	z = 2.02
	Missense	182.3	47	z = 4.90
	LoF	10.4	0	pLI = 0.97
	CNV	4.2	2	z = 0.41
**ZMYM2**	Synonymous	136.8	133	z = 0.20
	Missense	342	272	z = 1.85
	LoF	46.3	8	pLI = 0.97
	CNV	8.8	15	z = −0.59
**ZMYM4**	Synonymous	175.4	160	z = 0.72
	Missense	451.9	334	z = 2.71
	LoF	58.5	3	pLI = 1.00
	CNV	8.8	0	z = 1.53

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
