# Peer review of "The Diverging Routes of BORIS and CTCF: An Interactomic and Phylogenomic Analysis"

_life, 2018, doi:10.3390/life8010004_

Round 1

Reviewer 1 Report

CTCF is a ubiquitously expressed master regulator of genome architecture with a male germ-cell specific paralog, BORIS (also known as CTCFL). CTCF is conserved from Drosophila to man, while BORIS appears to have originated in the ancestor of amniotes. Here, the authors present a phylogenomic and interactomic analysis of CTCF and BORIS to better delineate their evolutionary trajectories. Using Cytoscape to query protein-protein interaction databases, the authors report the interactomes of CTCF and BORIS. The authors also report the presence of putative BORIS genes in a number of birds, in contrast to a prior study (Hore et al, PLOS Genet 2008). Integration of synteny analysis and chromosome conformation mapping data lead to a hypothesis that BORIS resides within a chromatin domain that includes other male germ cell-specific genes. Lastly, the authors examine ExAC data and find evidence that CTCF is less mutationally tolerant than BORIS.

Overall, the finding of BORIS in several birds is of interest, as is the synteny of the BORIS gene region and the differing mutational tolerance of CTCF and BORIS. However, I have significant concerns about the analysis of protein-protein interaction networks as presented, and the analysis of chromatin conformation raises some questions. My feeling is that a refocusing of the paper on the bird BORIS orthologs and the synteny of the BORIS region would greatly strengthen the central conclusions while avoiding more dubious hypotheses arising from the interactome analysis and, to a lesser extent, the Hi-C data.

-The finding of divergent protein-protein interaction networks for BORIS and CTCF is interesting but seems preliminary and incomplete. To the first point, no proteomic analyses of BORIS have been published; what has been published is limited to Y2H, co-IP, and other low-throughput biochemical assays. While I do appreciate that there is little overlap between their interaction partners at this stage, it seems there is insufficient evidence to make a broad conclusion. To the latter point, the Cytoscape queries are missing some interactors (though this is perhaps not the fault of the authors and instead reflects incomplete information in the queried databases). For instance, Rivero-Hinojosa et al (Sci Rep 2017) found a physical interaction between TAF7L and BORIS. Perhaps the most egregious omission is the interaction of CTCF and BORIS found in K562 cells (Pugacheva et al, Genome Biol 2015). The protein-protein interaction analysis is the weakest part of this paper, and I am uncertain of its value in the context of a phylogenomic study. At the very least, these database-excluded interactions should be mentioned, as they have important implications for CTCF and BORIS function. I think, overall, that the interactome analysis is fine presented in isolation, but its limitations should be more clearly spelled out and it should not be used to draw broad evolutionary inferences.

-I’m not convinced by the Hi-C analysis. While it does appear that SPO11, CTCFL, and PMEA1 are within a large loop of some sort, it looks like they each may further reside in sub-loops. This potentially undercuts the argument for co-regulation of these genes based on physical proximity in chromatin and should be further discussed/examined.

-Also related to the Hi-C data, what other genes are present in this larger loop? Based on the synteny blocks, there are genes between SPO11, CTCFL, and PMEA1. A quick look at the region in Genomicus and analysis of tissue-specific expression with the UCSC Genome Browser indicates that, for instance, the genes immediately adjacent to BORIS (PCK1 and RBM38) have rather broad expression patterns.

-Accession numbers for new BORIS orthologs should be reported.

-Accession numbers of Hi-C datasets used should be reported.

-The manuscript should undergo thorough language editing.

-Line 39: Suzuki et al (MCB 2010) should be cited as a study of genes dysregulated in BORIS-/- testes

-Line 47: Pugacheva et al showed that 29-38% of CTCF sites also have BORIS occupancy.

Author Response

Overall, the finding of BORIS in several birds is of interest, as is the synteny of the BORIS gene region and the differing mutational tolerance of CTCF and BORIS. However, I have significant concerns about the analysis of protein-protein interaction networks as presented, and the analysis of chromatin conformation raises some questions. My feeling is that a refocusing of the paper on the bird BORIS orthologs and the synteny of the BORIS region would greatly strengthen the central conclusions while avoiding more dubious hypotheses arising from the interactome analysis and, to a lesser extent, the Hi-C data.

 # We appreciate the reviewer's interest.  We also understood his/her concern with PPI network which we present differently in this new version of the manuscript.  We may on the contrary not share his/her feeling for its pertinence to the issue at stake, namely, PPI evolution after gene duplication.

#  We now simplify the network representation and correct  several mistakes. One major confusion  involved the gene H11 which is a heat shock protein.   If searched  using the UniProt  name convention (H11_HUMAN),  it  refers to  the Histone H1 variant. We corrected this mistake and  also added missing interactors mentioned below.

- The finding of divergent protein-protein interaction networks for BORIS and CTCF is interesting but seems preliminary and incomplete.

- No proteomic analyses of BORIS have been published; what has been published is limited to Y2H, co-IP, and other low-throughput biochemical assays. While I do appreciate that there is little overlap between their interaction partners at this stage, it seems there is insufficient evidence to make a broad conclusion.

# Even if incomplete, in our view, the poor overlap between the two networks is strong enough to draw the conclusion that they share  only few interaction partners.  T In addition, one would expect a smaller interaction network compared to CTCF from the limited expression of BORIS in  male germ cells , and one would expect a poor overlap of interaction networks from the high divergence rate of CTCFL's  amino-acid sequence (purifying selection weakened) after gene duplication.

- To the latter point, the Cytoscape queries are missing some interactors (though this is perhaps not the fault of the authors and instead reflects incomplete information in the queried databases). For instance, Rivero-Hinojosa et al (Sci Rep 2017) found a physical interaction between TAF7L and BORIS. Perhaps the most egregious omission is the interaction of CTCF and BORIS found in K562 cells (Pugacheva et al, Genome Biol 2015). The protein-protein interaction analysis is the weakest part of this paper, and I am uncertain of its value in the context of a phylogenomic study. At the very least, these database-excluded interactions should be mentioned, as they have important implications for CTCF and BORIS function. I think, overall, that the interactome analysis is fine presented in isolation, but its limitations should be more clearly spelled out and it should not be used to draw broad evolutionary inferences.

# We thank the reviewer for this comment.  We are in fact aware of  the work by Pugacheva et al., which we  quote in the context of competition for binding sites in sperm cells. In the revised manuscript, we  refer to the CTCF-BORS interaction and also cite the paper by Pugacheva et al. in this context.

# The comment raised by the reviewer about the missing TAF7L and CTCF interactors is due to the fact that these interactions are absent in the databases  used by  Cytoscape. We thank the reviewer for attracting our attention to this recent paper (Rivero-Hinojosa et al., Sci Rep 2017).  We added the TAF7L and CTCF/BORIS interaction in Table 1 and Fig. 2 and commented these observations.

-I’m not convinced by the Hi-C analysis. While it does appear that SPO11, CTCFL, and PMEA1 are within a large loop of some sort, it looks like they each may further reside in sub-loops. This potentially undercuts the argument for co-regulation of these genes based on physical proximity in chromatin and should be further discussed/examined.

 # We thank the reviewer and agree with  him/her  regarding a nested loop structure at this locus.  We have  reformulated  and conjecture  a possible interaction between the nested loops as a favourable conformation for co-regulation. At this stage, however, we can only formulate a model that needs to be corroborated or falsified  by additional experiments.

- Also related to the Hi-C data, what other genes are present in this larger loop? Based on the synteny blocks, there are genes between SPO11, CTCFL, and PMEA1. A quick look at the region in Genomicus and analysis of tissue-specific expression with the UCSC Genome Browser indicates that, for instance, the genes immediately adjacent to BORIS (PCK1 and RBM38) have rather broad expression patterns.

 # In the previous version,  we showed all the genes in Figure 4 , but we focused on these 3 genes (SPO11, CTCFL, and PMEA1).  Now we  present all the genes (seven in total) in the new figure and in the main text and added a supplementary table with more details. We also added an interpretation for the observed expression patterns in this genomic region (please see text). Our argument is based on the accepted definition of an insulated domain, or  regulatory unit.  Expression levels do not need to be similar (high or low) within such a domain as the distribution of regulatory  elements, such as enhancers, and their 3D vicinity in (sub)loops may also play critical roles.

-Accession numbers for new BORIS orthologs should be reported.

 # We provide now a supplementary table with Accession numbers.

-Accession numbers of Hi-C datasets used should be reported.

# The accessions are  now given in the material and methods section.

-The manuscript should undergo thorough language editing.

# Efforts in this direction have been made.

-Line 39: Suzuki et al (MCB 2010) should be cited as a study of genes dysregulated in BORIS-/- testes

 # We have added this reference

-Line 47: Pugacheva et al showed that 29-38% of CTCF sites also have BORIS occupancy.

#  This sentence is modified.

Reviewer 2 Report

Revision article: The diverging routes of BORIS and CTCF: An interactomic and phylogenomic analysis.

In the article authors describe their results obtained by bioinformatics analyses on the evolution of CTCF and its paralog CTCFL. They constructed the protein interaction networks,  highlighting that the networks are different, and studied the phylostratigraphic ages of interaction partners. Moreover, authors revealed that in nematodes the CTCF loss is accompanied by the loss of some interactors. Differently,  authors showed that CTCFL is lost in chicken but not from all birds. Then, authors studied the mutation tolerance for CTCF and CTCFL.

Despite the manuscript is interesting, not every results are convincing and well discussed. Therefore,  major revisions are necessary. 

In the figure 2 A, CTCF does not seem connected with H1.1. The comparison between interaction networks of CTCF and CTCFL show that only H2A connection is found in both networks. Therefore,  the sentences from “only one protein is found in both PPI networks, a sperm specific histone 1 variant (H1.1)” and subsequent discussion should be completely revised. 

Correspondingly,  Table 2 should be modified according to interaction networks and a new discussion about the interaction between CTCFL and the recent partner H1.1 should be performed.

Also in the discussion, this sentence and the subsequent should be revised: 

273 In this study, we compared protein interaction networks of CTCF and BORIS and identified an 

274 inflammasome component (IFI16), the small heat shock protein HSPB9 and the histone 1 variant H1.1 

275 as the only common interactors between CTCF and BORIS; these three proteins are the most recent 

276 interaction partners of CTCF/CTCFL and originated in the ancestor of mammals.

IFI16, HSPB9 and H1.1 are only the most recent interaction partners of CTCF.

A discussion on the interaction between  histone H3 isoforms and CTCFL could be interesting.

Minor revisions

Figure and table captions should be controlled and better specified.

Table 2: please, verify interactors and eliminate non human proteins. Distinguish CTCF (left) and CTCFL (right).

Figure 6: please highlight  coding and non coding variants and define “Non-coding transcript variant” histograms. Please better describe the normalization and define the axis.

Table 3 a and b: please render  uniform .

201 strong homology: two sequences, genes or proteins are homolog or not. Please substitute here and all over the text homology with similarity (the percentage of similarity or P values should be also indicated). 

Author Response

Despite the manuscript is interesting, not every results are convincing and well discussed. Therefore,  major revisions are necessary. 

- In the figure 2 A, CTCF does not seem connected with H1.1. The comparison between interaction networks of CTCF and CTCFL show that only H2A connection is found in both networks. Therefore,  the sentences from “only one protein is found in both PPI networks, a sperm specific histone 1 variant (H1.1)” and subsequent discussion should be completely revised. 

Correspondingly,  Table 2 should be modified according to interaction networks and a new discussion about the interaction between CTCFL and the recent partner H1.1 should be performed.

#  The network is now updated and the points mentioned by the reviewer are checked. . As pointed out in the comment to reviewer 1, we confused the gene H11 (Heat shock protein beta 8; alternative name: protein kinase H11) with the gene H1.1 (Histone H1.1). The main text is modified to correct these misunderstandings.

-Also in the discussion, this sentence and the subsequent should be revised: 

273 In this study, we compared protein interaction networks of CTCF and BORIS and identified an. 

274 inflammasome component (IFI16), the small heat shock protein HSPB9 and the histone 1 variant H1.1 

275 as the only common interactors between CTCF and BORIS; these three proteins are the most recent 

276 interaction partners of CTCF/CTCFL and originated in the ancestor of mammals.

IFI16, HSPB9 and H1.1 are only the most recent interaction partners of CTCF.

# This is done, see above.

A discussion on the interaction between  histone H3 isoforms and CTCFL could be interesting.

# We have mentioned the interaction with H3 isoforms and CTCFL and comment it shortly.

-Figure and table captions should be controlled and better specified.

Table 2: please, verify interactors and eliminate non human proteins. Distinguish CTCF (left) and CTCFL (right).

# We have controlled the tables, we especially took care of Table 2 and excluded mistakenly added accessions.

Figure 6: please highlight coding and non coding variants and define “Non-coding transcript variant” histograms. Please better describe the normalization and define the axis.

# This is done

Reviewer 3 Report

The authors have used well established methods to investigate the functional divergence between the two CCCTC-binding factors, CTCF and CTCFL, such as their genomic conservation of orthologous gene order (i.e., synteny) in vertebrates and their protein interaction networks.

I believe the authors have provided sufficient background information and explained the methods well. I also believe the authors presented the results well and withdrawn appropriate conclusions based on available data. I only have three minor issues that I could like to suggest to the authors:

1.     It would be helpful to provide some detailed explanations of the parameters used in the software used in the construction of the protein interaction network and annotation of network proteins.

2.     What is exactly an interactomic approach? Please explain.

3.     The two parts in Table 1 should be combined to make one table, currently Table 1 contains two tables.

Author Response

I believe the authors have provided sufficient background information and explained the methods well. I also believe the authors presented the results well and withdrawn appropriate conclusions based on available data. I only have three minor issues that I could like to suggest to the authors:

1.           It would be helpful to provide some detailed explanations of the parameters used in the software used in the construction of the protein interaction network and annotation of network proteins.

 # We have put some additional explanations for the interaction and annotation part.

 2.           What is exactly an interactomic approach? Please explain.

 # This is not a new terminology, although we agree that it is no widely used. It still occurs 315 times in PubMed from 2007 to 2017. Interactomics has been used to refer to the study of protein-protein interactions. The word "interactome" was originally coined in 1999 by a group of French scientists headed by Bernard Jacq (Sanchez et al., Nucleic Acids Res. 1999). In the framework of our study, we propose to extend this definition to the study of interaction as revealed by DNA conformation capture techniques.

 3.     The two parts in Table 1 should be combined to make one table, currently Table 1 contains two tables.

# We have reformatted Table 1 to meet the referee’s request.

Round 2

Reviewer 1 Report

The changes made by the authors have satisfactorily addressed by comments, and I think the paper is stronger now.

One minor addition: on lines 142-143, where the CTCF/BORIS and BORIS/TAF7L interactions are mentioned, Pugacheva et al (Genome Biol 2015) should be cited for CTCF/BORIS and Rivero-Hinojosa, Kang et al (Sci Rep 2017) should be cited for BORIS/TAF7L.

Author Response

One minor addition: on lines 142-143, where the CTCF/BORIS and BORIS/TAF7L interactions are mentioned, Pugacheva et al (Genome Biol 2015) should be cited for CTCF/BORIS and Rivero-Hinojosa, Kang et al (Sci Rep 2017) should be cited for BORIS/TAF7L.

# This is done.

Reviewer 2 Report

In the revised manuscript all concerns were taken into account and according changes were made.

The authors have satisfactorily answered the referees' requests.

The manuscript is now significantly improved.

I have just a suggestion.

In order to help the reader, the sentence:

152 There are 19 first-tier interaction partners of CTCFL, and

153 only three proteins are shared between both PPI networks, the heat shock protein beta-8, the histone

154 H2A type 2-C and BORIS.

could be modified adding acronyms after gene names and specifying the interactions CTCF-CTCF, CTCFL-CTCFL and CTCF-CTCFL, if they occur, because the fig.2 do not show them, differently from the table 2.

Author Response

# The requested changes are made.